# Metformin and Canagliflozin Are Equally Renoprotective in Diabetic Kidney Disease but Have No Synergistic Effect

**DOI:** 10.3390/ijms24109043

**Published:** 2023-05-20

**Authors:** Raphaëlle Corremans, Benjamin A. Vervaet, Geert Dams, Patrick C. D’Haese, Anja Verhulst

**Affiliations:** Laboratory of Pathophysiology, Department of Biomedical Sciences, University of Antwerp, 2610 Antwerp, Belgium; raphaelle.corremans@uantwerpen.be (R.C.); geert.dams@uantwerpen.be (G.D.); patrick.dhaese@uantwerpen.be (P.C.D.)

**Keywords:** chronic kidney disease, diabetic kidney disease, metformin, SGLT2-inhibitor

## Abstract

Diabetic Kidney Disease (DKD) is a major microvascular complication for diabetic patients and is the most common cause of chronic kidney disease (CKD) and end-stage renal disease. Antidiabetic drugs, such as metformin and canagliflozin, have been shown to exert renoprotective effects. Additionally, quercetin recently showed promising results for the treatment of DKD. However, the molecular pathways through which these drugs exert their renoprotective effects remain partly unknown. The current study compares the renoprotective potential of metformin, canagliflozin, metformin + canagliflozin, and quercetin in a preclinical rat model of DKD. By combining streptozotocin (STZ) and nicotinamide (NAD) with daily oral N(ω)-Nitro-L-Arginine Methyl Ester (L-NAME) administration, DKD was induced in male Wistar Rats. After two weeks, rats were assigned to five treatment groups, receiving vehicle, metformin, canagliflozin, metformin + canagliflozin, or quercetin for a period of 12 weeks by daily oral gavage. Non-diabetic vehicle-treated control rats were also included in this study. All rats in which diabetes was induced developed hyperglycemia, hyperfiltration, proteinuria, hypertension, renal tubular injury and interstitial fibrosis, confirming DKD. Metformin and canagliflozin, alone or together, exerted similar renoprotective actions and similar reductions in tubular injury and collagen accumulation. Renoprotective actions of canagliflozin correlated with reduced hyperglycemia, while metformin was able to exert these effects even in the absence of proper glycemic control. Gene expression revealed that the renoprotective pathways may be traced back to the NF-κB pathway. No protective effect was seen with quercetin. In this experimental model of DKD, metformin and canagliflozin were able to protect the kidney against DKD progression, albeit in a non-synergistic way. These renoprotective effects may be attributable to the inhibition of the NF-κB pathway.

## 1. Introduction

Diabetic kidney disease (DKD) is a major microvascular diabetic complication that takes place in 20% to 40% of all diabetic patients and is the leading cause of chronic kidney disease (CKD) worldwide [1]. DKD is a clinical syndrome predominantly caused by a long-standing disturbed glucose homeostasis and characterized by glomerular hyperfiltration, persistent albuminuria, and renal histological decay at glomerular, tubular, and interstitial level, which ultimately worsens renal functionality [2,3]. Careful control of blood glucose levels can help slow down or even prevent DKD [4]. In that regard, antidiabetic drugs play a prominent role in managing DKD. The US Food and Drug Administration (FDA) recently (2019) approved the use of sodium-glucose co-transporter-2 (SGLT-2) inhibitor, canagliflozin, for the treatment of DKD and very recently (2021) approved the use of dapagliflozin for patients with CKD irrespective of diabetes status [5,6,7,8]. Additionally, metformin, the first line treatment for type 2 diabetes, has shown effectiveness in decreasing all-cause mortality and delaying progression towards renal failure in DKD [9]. In addition, several preclinical studies have demonstrated the renoprotective effect of metformin in DKD by suppressing renal inflammation, oxidative stress, and fibrosis [10,11]. Furthermore, our lab already investigated the preventive and therapeutic efficacy of metformin in a nondiabetic setting. Metformin treatment was shown to strongly attenuate development of severe CKD, as well as being able to consistently halt progression of established CKD in rats [12,13]. Proteomic analyses revealed that metformin’s renoprotective effect was associated with the activation of the Hippo signaling pathway [12]. However, whether metformin is also able to halt the progression of DKD by altering the Hippo signaling pathway is not known. Yet, it has already been shown that the Hippo signaling pathway is inhibited under diabetic conditions, leading to yes-associated protein (YAP) translocation into the nucleus and thereby carrying its role in transcription [14]. Additionally, inhibition of the Hippo pathway accelerated the pathogenesis of DKD by promoting the proliferation of mesangial cells [14]. Furthermore, the Hippo pathway, and specifically the up-regulation of YAP, has also been linked to renal epithelial injury in the diabetic kidney [15]. Therefore, effective pharmacological interventions targeting the Hippo pathway, such as metformin, may slow down the progression of DKD. The renoprotective effect of SGLT-2 inhibitors is thought to be due to the suppression of hyperfiltration by restoring the tubuloglomerular feedback [16]. Since the pathophysiology of DKD consists of many abnormalities at the level of the kidney, combination therapy could be interesting. Additionally, recent evidence demonstrated that quercetin, a polyphenolic flavonoid, plays a significant role in reversing DKD by reducing oxidative stress [17]. Furthermore, Lei et al. demonstrated that quercetin delayed mesangial cell proliferation and alleviated renal function decline in diabetic mice through reactivation of the Hippo signaling pathway [18].

The present study aimed to compare the efficacy of metformin, canagliflozin, and a combination of both with quercetin with regards to the development and progression of DKD in rats. DKD was initiated by the induction of diabetes with streptozotocin (STZ) and nicotinamide (NAD), and combined with daily N(ω)-Nitro-L-Arginine Methyl Ester (L-NAME) administration, which further contributes to nephropathy development, as previously demonstrated by our lab [19].

## 2. Results

### 2.1. General Animal Aspects

Control animals significantly gained more weight compared to the diabetic rats from the start of the study until sacrifice. At week 9 and 12, the body weights of the canagliflozin- and metformin + canagliflozin-treated animals were significantly higher compared to the vehicle-treated animals (Figure 1A). Despite the reduced body weight of the diabetic rats, food intake was increased (Figure 1B). Additionally, water consumption and urinary volume were increased after diabetes induction (Figure 1C,D). Mortality was limited to five animals, spread over the different treatment groups (i.e., one in the vehicle-treated group, one in the metformin-treated group, one in the canagliflozin-treated group, and two in the quercetin-treated group).

### 2.2. Diabetes-Related Biochemical Parameters

The blood glucose levels of all diabetic animals were significantly increased at the start of the treatment compared to the control animals. From week 3 onwards until the end of the study, the blood glucose levels of the canagliflozin-treated and metformin + canagliflozin-treated animals were significantly lower compared to their baseline values before treatment, as well as compared to the vehicle-treated DKD rats at the same time points, which indicated the adequate blood glucose lowering effect of canagliflozin in the above mentioned groups (Figure 2A). Neither metformin, nor quercetin treatment was able to consistently reduce blood glucose levels in the diabetic rats. This was further confirmed by measuring the plasma glucose at week 12 (Figure 2D). Additionally, lactate levels were measured every 3 weeks. At the end of the study, lactate levels were significantly increased in all diabetic rats compared to their baseline values and control animals. However, with exception of a few individual animals, no value exceeded 5 mmol/L (Figure 2B). Furthermore, blood ketones, which also contribute to the acidification of the blood, were measured at the end of the study. Except for 2 quercetin-treated animals (>1.6 mmol/L), no onset of ketoacidosis was seen in the diabetic animals (Figure 2C).

### 2.3. Renal Function Is Affected by Diabetes Induction

As expected, creatinine clearance, as a measure for glomerular filtration rate (GFR), increased and manifested itself as hyperfiltration until the end of the study at week 12 (Figure 3A). The urinary albumin-to-creatinine ratio increased rapidly after diabetes induction. At the start of the treatment, albumin-to-creatinine values were at least five-fold higher compared to the non-diabetic control rats. This increase was sustained in all diabetic rats until the end of the study. In metformin (alone or in combination with canagliflozin)-treated animals, a trend toward lower albuminuria was noted compared to vehicle-treated rats (Figure 3B). Our preclinical model represents the early stage of DKD, in which an increment in serum creatinine was not yet seen (Figure 3C).

### 2.4. Metformin Ameliorates Urinary Markers of Kidney Damage

Urinary N-acetyl-β-D-glucosaminidase (NAG) can serve as a biomarker that increases with the severity of renal damage in DKD [20]. In our study, urinary NAG was increased in all diabetic animals at week 12 compared to non-diabetic control animals. However, when metformin was administered (alone or in combination with canagliflozin), urinary NAG was significantly lower compared to the vehicle-treated animals (Figure 4A). Furthermore, in our diabetic model, kidney injury molecule-1 (KIM-1), being an important biomarker in early DKD, was also increased [21]. Despite a similar trend, compared to NAG values, after metformin- and metformin + canagliflozin treatment, no significant reduction in KIM-1 level was observed (Figure 4B).

### 2.5. Blood Pressure

Two weeks after diabetes induction and L-NAME administration, i.e., at week 0 of the study, systolic blood pressure significantly increased in all DKD groups compared to the control rats. This increment was sustained until the end of the study (Table 1).

### 2.6. Metformin and Canagliflozin Both Halt Histological DKD Progression

The tubulointerstitial lesions observed in the diabetic animals were characterized by the ruffling and thickening of the tubular basement membrane, focal fibrotic regions, tubular epithelial cell necrosis, cell loss, tubular atrophy, tubular vacuolation, and tubular dilatation (Figure 5A). This phenotype was also present in the metformin-treated, canagliflozin-treated, and metformin + canagliflozin-treated animals, albeit in a noticeably milder form, as evidenced by the significant reduction in injured tubules with overt loss of epithelial cells (%), including cell death, cell loss, atrophy, and tubules with thickened basement membrane (Figure A1), in the aforementioned groups (Figure 5B). Masson’s Trichrome staining was performed to observe collagen accumulation (Figure 6A). The percentage of collagen in the vehicle- and quercetin-treated rats was significantly higher compared to non-diabetic control animals (Figure 6B). This was in contrast to metformin-, canagliflozin-, and metformin + canagliflozin-treated rats, which, compared to non-diabetic control rats, showed no increased collagen accumulation.

### 2.7. Metformin and Canagliflozin Inhibit NF-κB Signaling Pathway

To investigate the effect of the different treatments on fibrosis and inflammation, as well as the involvement of the Hippo signaling pathway in this pathology, the expression of a series of relevant genes was measured. In line with the Masson Trichrome-stained sections of the DKD rats, metformin-treated (alone or in combination with canagliflozin) animals showed a trend towards a diminished expression of the fibrotic gene, transforming growth factor β (Tgfβ) (Figure 7A), though this trend did not reach significance. The same was seen for the pro-inflammatory gene tumor necrosis factor α (Tnfα) (Figure 7B). Nonetheless, a significant reduction was observed in the gene expression of nuclear factor kappa light chain enhancer of activated B cells (NF-κB) and its downstream target Chemokine C-C motif -ligand 2 (Ccl2) after metformin or canagliflozin treatment (Figure 7C,D). When it comes to the Hippo pathway, mRNA analysis was not able to support the contribution of this pathway to DKD development, nor to the renoprotective effect of metformin and canagliflozin (Figure 7E–G).

## 3. Discussion

DKD is a major microvascular complication for diabetic patients and is the most common cause of CKD and end-stage renal disease. The current study compared the renoprotective effects of the antidiabetic drugs metformin and canagliflozin and the combination of both, during DKD development in rats. In addition, a comparison was made to treatment with quercetin, a natural flavonoid with reported renoprotection in other DKD and non-diabetic CKD models [18,22,23]. We report here that metformin and canagliflozin, but not quercetin, have promising renoprotective capacities (without synergism) during DKD development in rats. Moreover, we strengthen the potential of the NF-κB pathway as a therapeutic target.

DKD was induced in rats by STZ and NAD injections combined with L-NAME administration, as previously reported [19]. The strength of this model was reconfirmed by the fact that all rats developed severe hyperglycemia, indicating the induction of diabetes. Furthermore, as physiologically expected, diabetic animals drank more, had a higher urinary output, and ate more while gaining less weight compared to the non-diabetic control animals. Remarkably, while chronic hyperglycemia was observed in the quercetin-treated DKD rats, the animals had a lower water and food intake, as well as lower urinary volume. One would actually expect higher water intake and increased urinary volume with hyperglycemia or, at least, a glucose lowering effect of quercetin. Perhaps a certain degree of physiological divergence associated with physical and physiological disturbance may have affected the expected responses. In accordance with this, we should note that, at the end of the study, the quercetin-treated rats were rather weak, which might have interfered with appetite, whereas lower urinary volume (hence water intake) may have come from a certain degree of malfunctioning water regulation due to injured kidneys. Blood pressure remained significantly higher, attributable to L-NAME administration, until the end of the study. Surprisingly, the quercetin-treated animals showed decreased blood pressure at the end of the experiment. Several studies performed in humans and animals, including a very recent study by Popiolek-Kalisz et al. [24], describe that quercetin supplementation is able to decrease blood pressure in normotensive and (pre)hypertensive patients. Unfortunately, the mechanisms by which quercetin affects blood pressure remain partly unclear [24]. Different mechanisms of action have been proposed, including the following: reduction in oxidative stress, interference with the RAAS system (specifically ACE inhibitors), and /or improvements in endothelial and/or vascular function [25]. Furthermore, in accordance with the inherent features of diabetes and CKD [26], blood lactate levels showed a chronic increment in all diabetic groups. Importantly, none of the treatments (incl. metformin) aggravated lactate buildup. On a functional and histopathological level, DKD development was characterized by hyperfiltration and proteinuria, as well as the presence of injured tubules and interstitial fibrosis. In concordance with the histological damage, the tubular damage markers, urinary NAG, and KIM-1 levels were increased. Urinary NAG is considered as a good biomarker for the diagnosis of diabetic nephropathy [27]. Additionally, Omozee et al. demonstrated that the increase in urinary NAG concentration preceded the increase in albumin excretion, which implies that, even before microalbuminuria begins to appear in urine, urinary NAG levels are already increased [27]. In our study, we observed significantly increased NAG levels in the urine of vehicle-treated diabetic rats compared to the control animals, indicating the induction of DKD. Although serum creatinine was not significantly increased, hyperfiltration and proteinuria were present after diabetes induction, as evidenced by the increased creatinine clearance and albumin-to-creatinine ratio. Overall, this is indicative of the onset of DKD with clear evidence of subclinical injury.

With regard to the effects of separate metformin and canagliflozin treatments, we can conclude that both drugs attenuated DKD development to a similar extent since both compounds equally tempered histopathological renal damage. It is known that glycemic control is an important factor in halting DKD progression [28]. The effect of canagliflozin is clearly associated with a reduced blood glucose level, while this is not the case for the effect of metformin. Indeed, metformin was not able to reduce blood glucose from week 6 onwards until the end of the study at week 12, which is in contrast to canagliflozin. Additionally, the histopathological effect of metformin (alone or in combination with canagliflozin) could be further confirmed by the significantly reduced urinary excretion of NAG. Since this lysosomal enzyme is present in renal proximal tubular cells, reduced urinary excretion of NAG, compared to vehicle-treated animals, indicates less renal tubular dysfunction and suggests an alleviation of renal injury in the animals treated with metformin [29,30]. The same trend was seen in urinary KIM-1 excretion and albuminuria at week 12. KIM-1 is a transmembrane glycoprotein present in proximal tubular cells and used as urinary biomarker of kidney damage [31,32].

Given the previous findings in our lab, the protective effect of metformin in non-diabetic CKD animals was attributed to the activation of the Hippo signaling pathway [12]. Interestingly, in diabetic db/db mice, Lei et al. demonstrated that quercetin, a bioactive polyphenolic flavonoid with antioxidant properties, reactivated the Hippo signaling pathway and alleviated the issues of both renal function decline and inhibited mesangial cell proliferation [18,33]. Furthermore, Lai et al. demonstrated that the positive effect of quercetin on renal function was attributable to the inhibition of the overexpression of Tgfβ1 and connective tissue growth factor (Ctgf) in STZ-induced diabetic rats, the latter being directly related to the Hippo pathway [23]. However, in our current study, the pathophysiological disturbances, induced by DKD, were not ameliorated by the administration of quercetin. In addition, only one (Ctgf) out of three Hippo-pathway related genes was up-regulated upon treatment with this drug. Overall, this indicates that the consistent involvement of the Hippo signaling pathway regulation in DKD might have to be nuanced and is, at least, model dependent.

Metformin and canagliflozin treatment were able to significantly reduce mRNA expression of NF-κB. Since a directly proportional relationship between NF-κB mRNA, protein expression, and even activity was found in multiple studies, we speculate that the underlying molecular mechanism responsible for the renoprotective effect of these drugs may be attributable to a reduced inflammatory response in this preclinical model (Figure 8) [34,35]. Indeed, NF-κB is a DNA binding protein factor that serves as a pivotal mediator of inflammatory responses [36,37]. This is also in line with the down-regulated mRNA expression of Ccl2, a downstream target of NF-κB and emerged as a key mediator of inflammation in DKD [38,39]. The NF-κB pathway responds to diverse stimuli. Inhibition of these stimuli by metformin (reducing TNFα expression) and canagliflozin (reducing hyperglycemia) may diminish the pathogenic process of inflammation. Furthermore, by inhibiting the inducible degradation of the NF-κB inhibitor protein α (IκBα), triggered by the phosphorylation of IκB kinase (IKK), the NF-κB pathway might also be suppressed (Figure 8). Likewise, Hattori et al. showed that metformin was able to inhibit TNFα-induced NF-κB activation by suppressing IKK activity and inhibitory protein κBα (IκBα) phosphorylation in vascular endothelial cells [40]. By doing so, the NF-κB dimers are not released, and the inflammatory process is suppressed (see Figure 8). This is in line with a series of experimental studies that investigated the beneficial impact of metformin in the pathological process of DKD [41]. Al Za’abi et al. suggested that metformin could attenuate the progression of CKD in both diabetic and non-diabetic rats by alleviating inflammation and apoptosis through the inhibition of NF-κB signaling [42]. Additionally, Zhang et al. revealed that the renoprotective effect of metformin against the progression of DKD in high-fat diet and STZ-induced diabetic rats is attributable to its anti-inflammatory and anti-oxidative functions [43]. Additionally Sun et al. concluded that metformin is promising for the treatment of DKD since metformin was able to inhibit the inflammatory response, ROS-mediated cell apoptosis, and fibrosis in transgenic db/db mice [44]. With regards to canagliflozin, Vallon et al. suggested that a reduction in renal mRNA content of NF-κB in diabetic Akita mice was caused by SGLT2 inhibition and appeared to depend on blood glucose reduction, which seems to be in line with our results [45].

Since the known molecular mechanisms by which metformin (insulin sensitizer) and canagliflozin (causing glucosuria) exert their glucose lowering effects are completely different, a possible synergistic effect on DKD development could not be excluded [46]. However, our study did not show a synergetic positive effect of co-administering both compounds. This is evidenced by a similar reduction in (i) the percentage of injured tubules with an overt loss of epithelial cells (including cell death, cell loss, atrophy, and tubules with thickened basement membrane), (ii) collagen accumulation, and (iii) NF-κB mRNA levels, as compared to monotherapy of the compounds.

The current work has some limitations. First, gene expression was measured; however, this does not necessarily reflect the proteins expression levels and hence may provide only partial insight into pathway involvement. Second, the kidney injury pattern demonstrated mild glomerular lesions relative to the more severe tubulointerstitial damage. The fact that the terminal stage of glomerulosclerosis, as seen in human DKD, is not observed in our study might be considered a limitation of this model. However, in accordance with our findings, Di Vincenzo et al. [47] and Fioretto et al. [48] also noted an absence of pronounced glomerulopathy in kidney biopsies of patients with type 2 diabetes, whilst tubulointerstitial abnormalities occurred more regularly [47,48].

## 4. Materials and Methods

### 4.1. Drug Preparation

STZ and NAD were dissolved in physiological water at concentrations of 65 mg/mL and 115 mg/mL, respectively. STZ was injected intravenously, and NAD was injected intraperitoneally. L-NAME was dissolved in 1% carboxymethylcellulose at a concentration of 10 mg/mL before daily oral gavage. In addition, metformin, and/or canagliflozin, or quercetin were added in the 1% carboxymethylcellulose/L-NAME solution at concentrations of 20 mg/mL, 2.5 mg/mL, and 5 mg/mL, respectively, before daily oral gavage. A dose volume of 10 mL/kg was used.

### 4.2. Animal Experiment

Seventy-eight (78) male Wistar rats were housed (two per cage) and exposed to a 12–12 h light–dark cycle, with free access to water and a standard diet. Diabetes was induced as previously described [19]. Two weeks after diabetes induction and L-NAME administration, diabetic rats were randomly assigned to five different treatment groups to receive either vehicle, metformin, canagliflozin, metformin + canagliflozin, or quercetin in combination with L-NAME for a period of 12 weeks. As a study reference, a group of vehicle-treated non-diabetic control rats with normal renal function was also included (Figure 9). Every 3 weeks, animals were housed in metabolic cages for 24 h to collect urine samples, followed by blood sampling through the tail vein in a restrained, conscious condition. Animals were sacrificed by exsanguination through the retro-orbital plexus after anesthesia through the intraperitoneal administration of 60 mg/kg ketamine and 7.5 mg/kg xylazine. Experimental procedures were conducted according to the National Institutes of Health Guide for the care and Use of Laboratory Animals 85-23 (1996) and approved by the University of Antwerp Ethics Committee (permit number 2020-55).

### 4.3. Biochemical Analyses

Serum creatinine was measured using an Atellica CH 930 Analyzer (Siemens Heathineers, Erlangen, Germany). To assess renal function, creatinine was also measured in urine samples according to the Jaffe principle. Urinary albumin concentration was measured using Hemocue^®^Albumin 201 System (HemoCue AB, Ängelholm, Sweden). Blood glucose (GlucoMen^®^ Aero, A. Menarini diagnostics, Machelen, Belgium) and lactate levels (StatStrip Xpress, Nova Biomedical, Den Bosch, The Netherlands) were determined every 3 weeks until sacrifice. Plasma glucose was measured by using a colorimetric Assay Kit (Sigma-Aldrich, Saint Louis, MO, USA). At sacrifice, blood β-ketone levels were measured (GlucoMen^®^ Aero 2K, A. Menarini diagnostics, Machelen, Belgium). Urinary NAG (Roche Diagnostics GmbH, Mannheim, Germany) and KIM-1 (MyBioSource, San Diego, CA, USA) were determined photometrically at the end of the study.

### 4.4. Blood Pressure

The blood pressure of restrained awake rats was non-invasively measured at baseline (week 0), week 6, and before sacrifice (week 12) using a tail-cuff system (CODA system, Kent Scientific, Torrington, CT, USA).

### 4.5. Histology

The left kidney was isolated and decapsulated, after which a transverse slice was fixed (4 h) in neutral buffered formalin, rinsed with isopropanol, and embedded in paraffin. General morphological evaluation and quantification of the percentage of tubular lesions and the severity of interstitial lesions was performed on 4 µm-thick sections that were stained with Periodic acid–Schiff (PAS). Four random pictures (200×) were taken of the cortex of each PAS-stained renal section. Using Fiji imageJ 1.53 analysis software, the area that contained injured (cell death/cell loss/atrophy) tubules was measured. The ratio of the injured tubules with overt loss of epithelial cells area to the total area resulted in the percentage of area taken by the injured tubules, represented by tubules subjected to cell death, as well as cell loss, atrophy, and tubules with thickened basement membrane (Figure A1). Masson’s staining was performed to evaluate collagen expression. Four random pictures (200×) were taken of the cortex of each Masson-stained renal section and quantified using color threshold analysis in Fiji image analysis software.

### 4.6. Quantitative Real-Time PCR

Total mRNA of a snap-frozen kidney section was extracted using the PureLink^®^ RNA Mini Kit (Invitrogen by Thermo Fischer Scientific, Waltham, USA) and reverse transcribed to cDNA using the High-Capacity cDNA archive kit (Applied Biosystems by Thermo Fischer Scientific, Waltham, USA). Quantitative real-time polymerase chain reaction (qPCR) was performed using QuantStudio 3 (Thermo Fischer Scientific, Waltham, USA). TaqMan probe and primers were purchased from Thermo Fischer for glyceraldehyde 3-phosphate dehydrogenase (Gapdh) (Rn99999916_s1), transforming growth factor beta (Tgfβ) (Rn00572010_m1), tumor necrosis factor alpha (Tnfα) (Rn00562055_m1), nuclear factor kappa light chain enhancer of activated B cells (NF- κB) (Rn01399572_m1), chemokine C-C motif -ligand 2 (Ccl2) (Rn00580555_m1), yes-associated protein 1 (Yap1) (Rn01448051_m1), connective tissue growth factor (Ctgf) (Rn01537279_g1), and cysteine-rich angiogenic inducer 61 (Cyr61) (Rn00580055_m1). The expression level of each tested gene was analyzed in triplicate for each sample and normalized to the expression of the housekeeping reference gene Gapdh.

### 4.7. Statistics

The results are expressed as means ± SEM. To investigate the statistical differences between groups a Kruskal–Wallis test followed by Mann–Whitney U test was applied. Comparisons between timepoints within each group were performed with a Friedman test followed by Wilcoxon signed-rank test. All statistical tests were performed using Prism 9 (GraphPad Software, San Diego, CA, USA). A *p*-value of <0.05 was considered statistically significant.

## 5. Conclusions

In conclusion, metformin was able to protect renal proximal tubular cells, reduce injured tubules, and prevent collagen accumulation, even in the absence of proper glycemic control. On the other hand, canagliflozin showed some promising results, which are most likely attributable to its capacity to lower blood glucose during the study. This is consistent with our previous study, in which the absence of a renoprotective effect of canagliflozin was observed in a non-diabetic CKD model [12]. Quercetin did not affect the progression of DKD in our study, thereby indicating that the Hippo signaling pathway does not seem to play a major role in DKD development or progression in this model. Alternatively, the NF-κB pathway may, at least in part, be more important since its down-regulation, as well as the down-regulation of its downstream target Ccl2, was associated with the renoprotective actions of metformin and canagliflozin in DKD development. Further fundamental research is necessary to elucidate the complex network of signaling pathways by which metformin and canagliflozin exert their renoprotective effects in DKD, as this is indispensable to identify potentially new therapeutic targets.

## Figures and Tables

**Figure 1 ijms-24-09043-f001:**
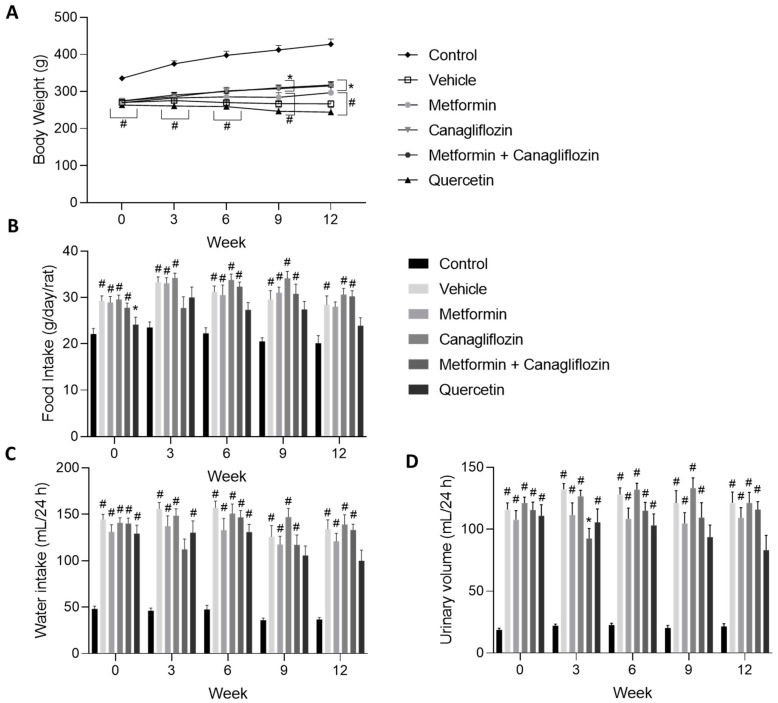
Changes in body weight (**A**), food consumption (**B**), water intake (**C**), and urinary volume (**D**) in diabetic rats treated with vehicle, metformin, canagliflozin, metformin + canagliflozin, or quercetin for a period 12 weeks. Data are presented as mean ± SEM. * *p* < 0.05 vs. vehicle-treated diabetic rats, # *p* < 0.05 vs. non-diabetic control animals with normal renal function.

**Figure 2 ijms-24-09043-f002:**
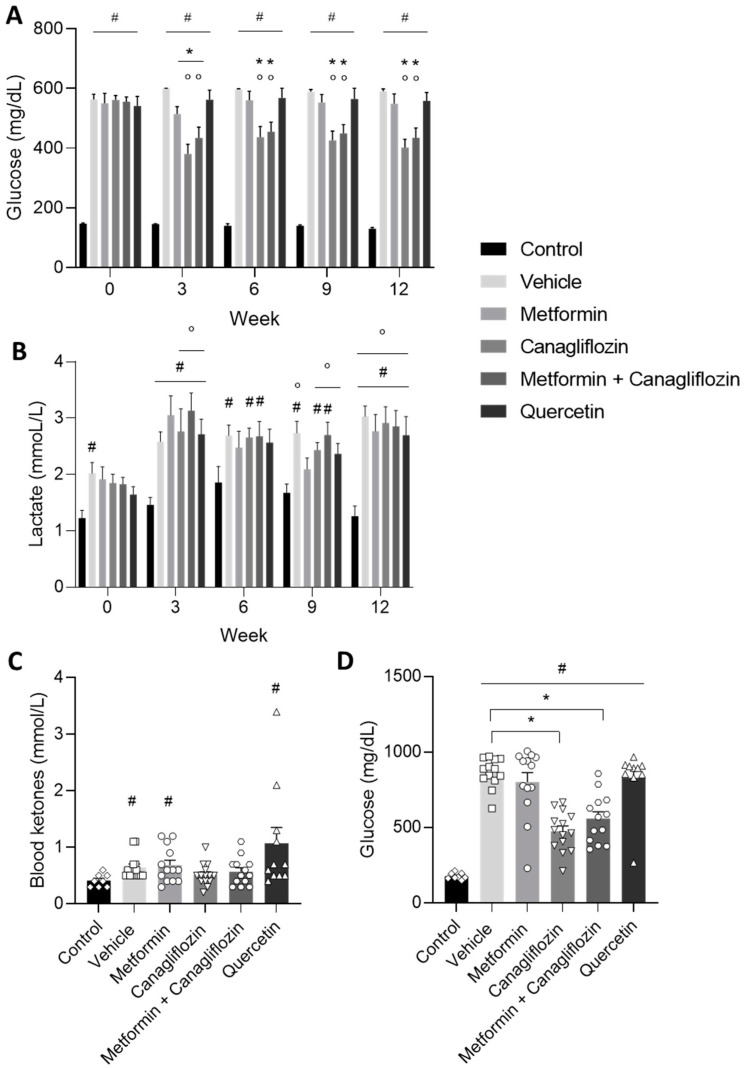
Blood glucose (**A**), lactate (**B**), ketones (week 12) (**C**), and plasma glucose (week 12) (**D**) levels in non-diabetic control rats and in diabetic rats treated with vehicle, metformin, canagliflozin, metformin + canagliflozin, or quercetin for a period of 12 weeks. Blood glucose test range: 20–600 mg/dL. Data are presented as mean ± SEM. * *p* < 0.05 vs. vehicle-treated diabetic rats, # *p* < 0.05 vs. non-diabetic control rats with normal renal function and ° *p* < 0.05 vs. week 0.

**Figure 3 ijms-24-09043-f003:**
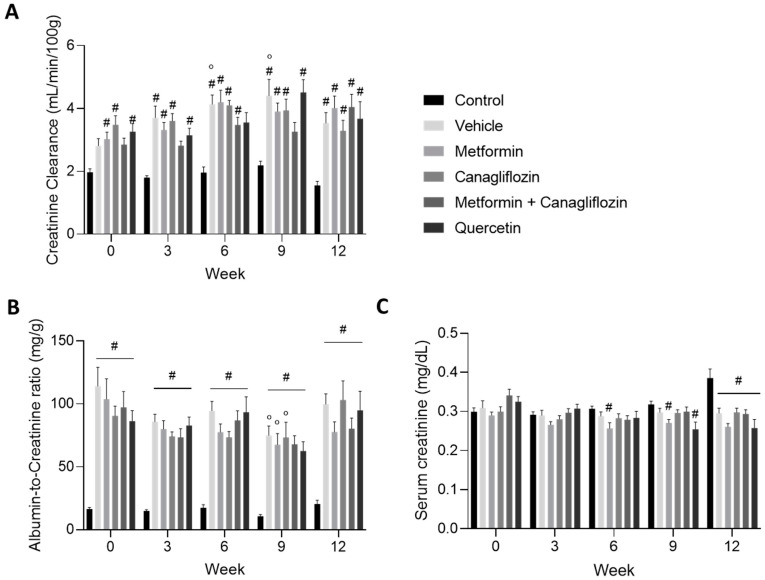
Creatinine clearance (**A**), and urinary albumin-to-creatinine ratio (**B**), serum creatinine (**C**), in non-diabetic control rats and in diabetic rats treated with vehicle, metformin, canagliflozin, metformin + canagliflozin, or quercetin for a period of 12 weeks. Data are presented as mean ± SEM. # *p* < 0.05 vs. non-diabetic control rats with normal renal function and ° *p* < 0.05 vs. week 0.

**Figure 4 ijms-24-09043-f004:**
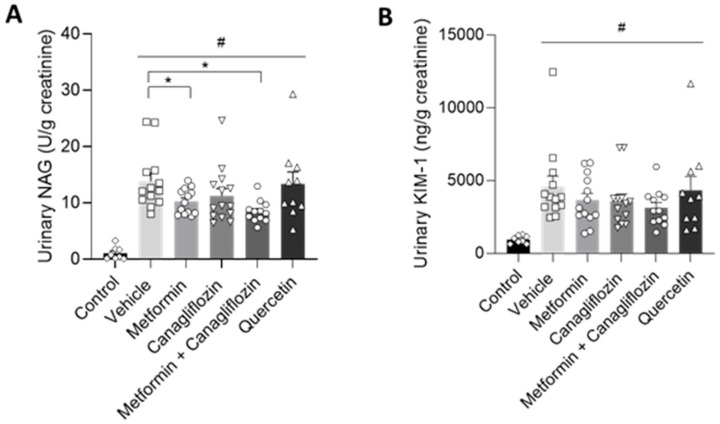
Alterations in urinary NAG (**A**) and KIM-1 (**B**) levels at the end of the study in non-diabetic control rats and in diabetic rats treated with vehicle, metformin, canagliflozin, metformin + canagliflozin, or quercetin for a period of 12 weeks. Data are presented as mean ± SEM. # *p* < 0.05 vs. non-diabetic control rats and * *p* < 0.05 vs. vehicle-treated diabetic rats.

**Figure 5 ijms-24-09043-f005:**
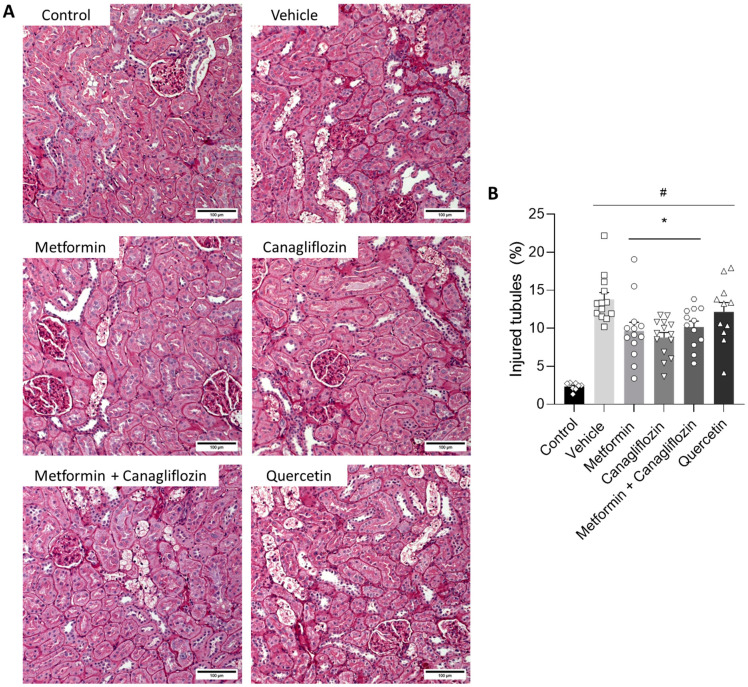
Effect of metformin, canagliflozin, metformin + canagliflozin, and quercetin on injured tubules with overt loss of epithelial cells (%). Periodic acid–Schiff-stained renal sections (200×) (**A**) injured tubules with overt loss of epithelial cells area percentage (**B**) in non-diabetic control rats and diabetic rats. Data are represented as mean ± SEM. Bars = 100 µm. * *p* < 0.05 vs. vehicle-treated diabetic rats, # *p* < 0.05 vs. non-diabetic control rats with normal renal function.

**Figure 6 ijms-24-09043-f006:**
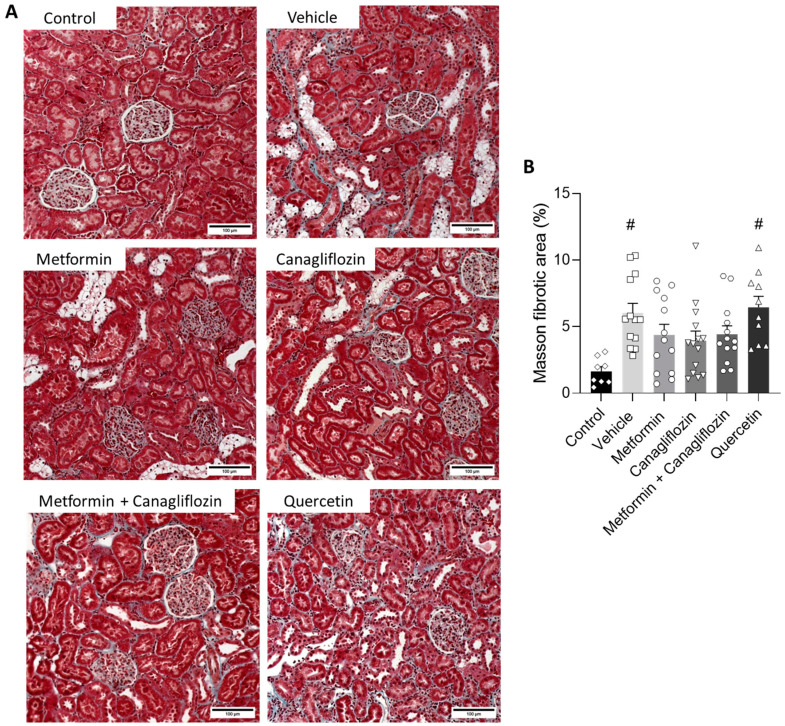
Effect of metformin, canagliflozin, metformin + canagliflozin, and quercetin on fibrosis. Masson trichrome-stained renal sections (200×) (**A**) and Masson’s fibrotic area percentage (**B**) in non-diabetic control rats and diabetic rats. Data are represented as mean ± SEM. Bars = 100 µm. # *p* < 0.05 vs. non-diabetic control rats.

**Figure 7 ijms-24-09043-f007:**
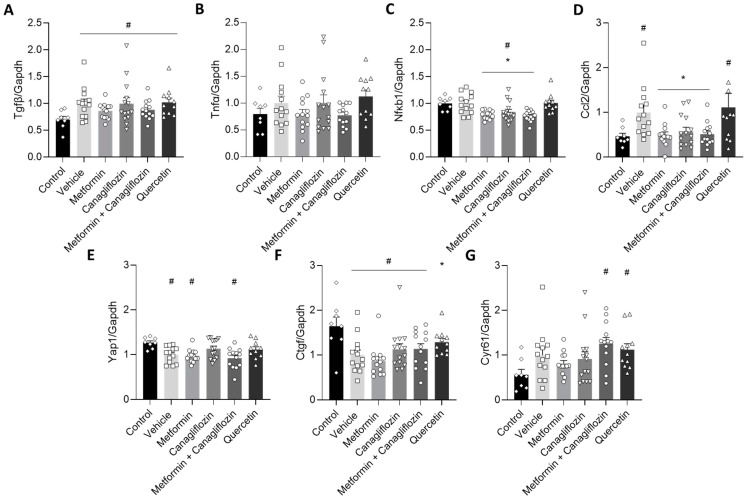
Gene expression analysis of the effect of different treatments on fibrosis (Tgfβ) (**A**), inflammation (Tnfα) (**B**), the NF-κB pathway (**C**) and its downstream target (Ccl2) (**D**), as well as the Hippo pathway (Yap1) (**E**) and its downstream targets (Ctgf and Cyr61) (**F**,**G**). Data are represented as mean ± SEM. # *p* < 0.05 vs. non-diabetic control rats with normal renal function, * *p* < 0.05 vs. vehicle-treated diabetic rats.

**Figure 8 ijms-24-09043-f008:**
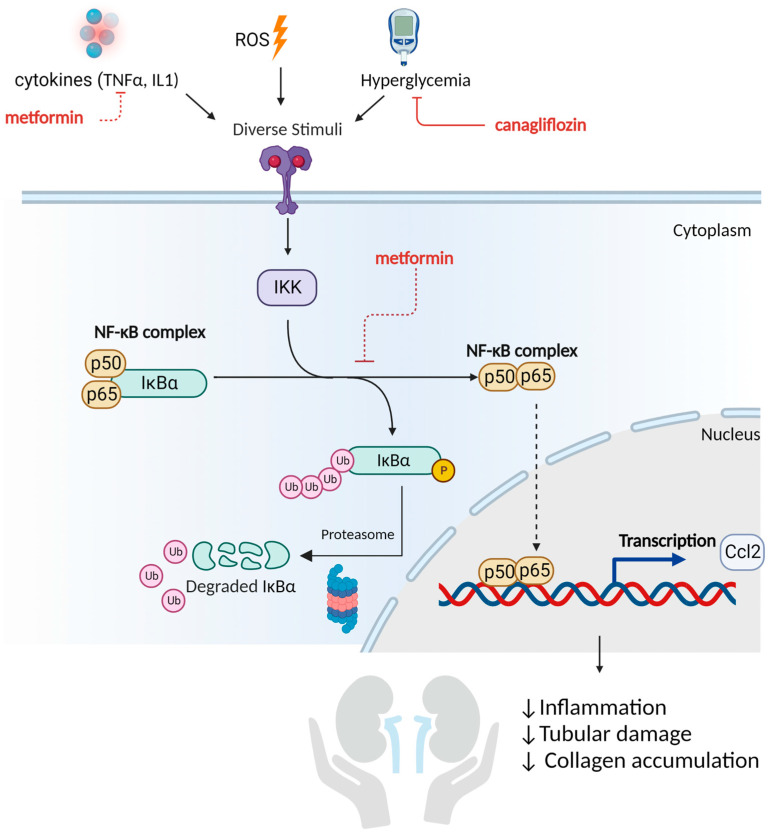
Reduced activation of the NF-κB pathway as a mediator of metformin’s and canagliflozin’s renoprotective effect. The nuclear factor κB (NF-κB) pathway plays a central role in the pathogenesis and progression of DKD. In its inactive state, the NF- κB complex is located in the cytoplasm bound with inhibitory protein κBα (IκBα). Various stimuli can activate the IκB kinase (IKK). After activation, IKK phosphorylates the IκB protein, which results in its ubiquitination and degradation by the proteasome. This releases the NF-κB dimers (p50-p65) from negative regulation. The activated NF-κB complex is then translocated into the nucleus to initiate gene transcription of, e.g., cytokines such as Ccl2. The anti-inflammatory activity of metformin is attributable to its capacity to suppress IKK activity/IκBα phosphorylation and reduce TNFα, while the anti-inflammatory effect of canagliflozin is due to the suppression of hyperglycemia. Created with Biorender.

**Figure 9 ijms-24-09043-f009:**
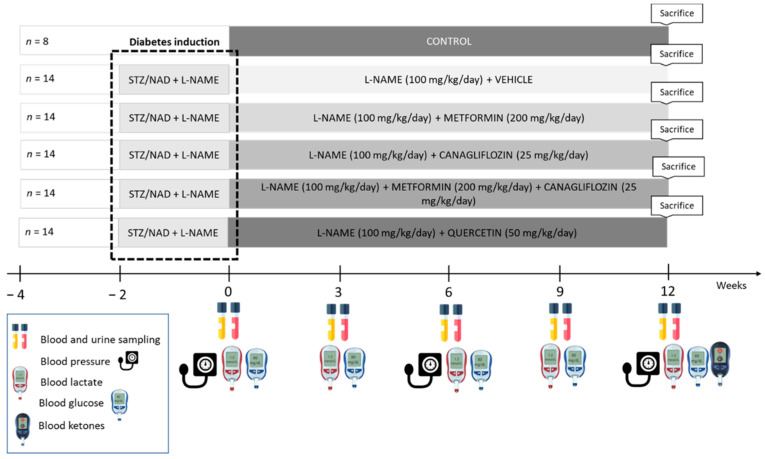
Study setup. STZ, streptozotocin; NAD, nicotinamide; L-NAME, N(ω)-Nitro-L-Arginine Methyl Ester.

**Table 1 ijms-24-09043-t001:** Systolic and diastolic blood pressure of non-diabetic control rats and diabetic rats treated with vehicle, metformin, canagliflozin, metformin + canagliflozin, or quercetin.

Blood Pressure	Systolic Blood Pressure (mm Hg)			Diastolic Blood Pressure (mm Hg)		
Week	0	6	12	0	6	12
Control	134 ± 7	134 ± 11	140 ± 7	93 ± 8	95 ± 4	100 ± 9
Vehicle	172 ± 7 #	165 ± 7 #	175 ± 5 #	128 ± 8 #	125 ± 7 #	128 ± 5 #
Metformin	170 ± 6 #	175 ± 7 #	168 ± 10 #	128 ± 6 #	130 ± 7 #	131 ± 8 #
Canagliflozin	166 ± 7 #	184 ± 6 #°	186 ± 7 #°	124 ± 6 #	143 ± 6 #°	138 ± 7 #
Metformin + Canagliflozin	170 ± 7 #	177 ± 6 #	183 ± 8 #	119 ± 12	134 ± 6 #	141 ± 8 #
Quercetin	166 ± 8 #	171 ± 7 #	149 ± 8 *	128 ± 8 #	125 ± 8 #	104 ± 6 °*

# *p* < 0.05 vs. Control; * *p* < 0.05 vs. Vehicle; ° *p* < 0.05 vs. week 0.

## Data Availability

The data presented in this article are available upon request from the corresponding authors.

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
