# Peer review of "Metformin and Canagliflozin Are Equally Renoprotective in Diabetic Kidney Disease but Have No Synergistic Effect"

_ijms, 2023, doi:10.3390/ijms24109043_

Round 1

Reviewer 1 Report

The subject of this manuscript is of interest. The use of pcr validation mechanisms alone in experiments is not sufficient, and attention should be paid to abbreviations.

Reviewer 2 Report

The publication describes the clinically significant use of a new group of antidiabetic drugs (like canagliflozin) as well as metformin and quercetin - in an animal model as a therapy that delays the development of chronic kidney failure in the course of diabetes. This model indicates the possibility of using these drugs (metformin and canagliflozin) interchangeably (there was no synergistic effect) in preventing diabetic kidney damage. The references cited by the authors are adequate for the content. The text is clear and concise and the methods are described correctly. I would only suggest improving the readability of Figures 4, 7 (entire), and 4c.

Author Response

Dear Reviewer,

Based on your constructive comments, we wish to present you a revised version of the manuscript (ijms-2236941) entitled ‘Metformin and canagliflozin are equally renoprotective in diabetic kidney disease but have no synergistic effect’ for your consideration to be published in the International Journal of Molecular Sciences, and more precisely in the special issue ‘Diabetic Kidney Diseases’.

We thank you for the careful reading of our work. We have implemented your suggestion to improve  the readability of the figures.  

Reviewer 3 Report

Corremans et al performed a study to compare the renoprotective effects of routinely used antidiabetic drugs, metformin and canagliflozin in a newly established rat model of diabetic kidney disease. The study is well designed, and the manuscript is well written but my major concern is that the conclusion does not match the data presented. Majority of the data were not statistically significant, thus discussion and conclusion need to be modified to reflect the data.

Major comment:

1. The significant findings presented in the paper are metformin moderately reduced an early marker of diabetic kidney disease, urinary NAG, and both metformin and canagliflozin reduced the percentage of injured tubules in the diabetic kidneys.

However, the more widely used markers of renal dysfunction eg albuminuria and KIM-1 are not significantly changed, although there is a trend towards improvement. So to conclude that metformin and canagliflozin are renoprotective (but non-synergistic) during DKD development in rats (line 193-194)” is a little ambitious.

2. The authors also conclude that NF-κB pathway is a target of metformin and canagliflozin in DKD. However, NF-kB expression was not changed in diabetes in the study, thus the significant reduction in NF-kB expression when compared to control might not be a good thing. Gene expression also does not reflect its activity. Is NF-kB activity changed? Downstream marker tnfa expression does not seem to change, so the question is, is NF-kB activated in this model? What about other downstream markers of NF-kB eg IL-6, MCP1 etc? Protein expression might be a better indicator than gene expression.

3. The method used to quantitate injured tubules is very ambiguous. Arrows or asterisks in the image will make it useful to distinguish “thickening of the tubular basement membrane, focal fibrotic regions, tubular epithelial cell necrosis, tubular atrophy, tubular vacuolation and tubular dilatation” line 148-149. However, the quantitation presented only takes into account tubular cell death (methods line 328-331). In addition, the marker of tubular injury, urinary KIM-1 does not reflect a change in tubular injury in the treatment groups. Additional experiments are necessary to validate this result.

4. What about glomerulosclerosis which usually occurs before tubular injury in preclinical models of diabetic kidney disease? This can be assessed by PAS staining and either GSI scoring or % of mesangial area.

5. The representative images in Figure 6 do not represent the graphs as there is hardly any positive blue staining. The authors could use picrosirius red or collagen I immunohistochemistry to validate tubulointerstitial fibrosis in this model.

6. I think the involvement of Hippo pathway is interesting but the authors were quick to dismiss its association. I would suggest to either not mention it in introduction or delve deeper and look at protein expression of downstream markers. Gene expression is often timepoint related and might not tell the full story.

7. Can the authors comment why quercetin is not renoprotective in this model, but others have shown differently? doi: 10.1039/d1fo03958j

Minor comments

1. A reference is require for the sentence “Proteomic analyses revealed that metformin’s renoprotective effect was associated with the activation of the Hippo signaling pathway (line 50-53).”

2. The authors mentioned there were 5 premature deaths (line 82) but from which group? Is it treatment related?

4. Fig1A – cannot differentiate between canagliflozin and metformin+canagliflozin groups. Change to bar charts like the rest.

5. Blood glucose – glucometer maximum detection is 600mg/dL so this is not a true representative of data. Need plasma glucose measurement. What about HbA1c?

6. Can the authors explain why in the quercetin-treated group, blood glucose was high but the rats had lower water intake and urine volume? Also, how does quercetin affects blood pressure and the significance, if any, of this result.

7. Markers of inflammation should be examined (Figure 8).

Round 2

Reviewer 1 Report

  • After revision, the manuscript was considered ready for publication

Author Response

Thank you for your revision.   

Reviewer 3 Report

I appreciate the authors’ careful and detailed response to my comments. However, majority of the comments were not incorporated into the revised manuscript and I do not think this has improved the quality or significance of the manuscript considerably. I understand this could be partly due to the short timeframe provided by the journal/editor, as alluded by the authors, but this should not impede the quality of publishable work.

 1. Plasma glucose (not blood) can be measured using simple colorimetric assay kit with or without sodiumfluoride.

2. Many of the explanations are provided in the cover letter but not incorporated into the manuscript.

3. GSI and other fibrotic/inflammatory markers should be included to strengthen the findings.

4. There should also be more exploration of mechanisms ie NFkB or other pathways.

Author Response

Sincerly yours,

Raphaëlle Corremans

Round 3

Reviewer 3 Report

All comments have been adequately addressed.